# Assessing Asymptomatic Malaria Carriage of *Plasmodium falciparum* and Non-*falciparum* Species in Children Resident in Nkolbisson, Yaoundé, Cameroon

**DOI:** 10.3390/children8110960

**Published:** 2021-10-24

**Authors:** Nji Mbuh Akindeh, Lesley Ngum Ngum, Peter Thelma Ngwa Niba, Innocent Mbulli Ali, Ornella Laetitia Oben Ayem, Jean Paul Kengne Chedjou, Calvino Tah Fomboh, Aristid Herve Mbange Ekollo, Cyrille Mbanwi Mbu’u, Wilfred Fon Mbacham

**Affiliations:** 1Biotechnology Center, University of Yaoundé I, Messa, Yaoundé P.O. Box 3851, Cameroon; ngumngumlesley@gmail.com (L.N.N.); thelma2009@yahoo.co.uk (P.T.N.N.); innocent.mbulli@univ-dschang.org (I.M.A.); obenlaetitia@gmail.com (O.L.O.A.); chedjouj@yahoo.fr (J.P.K.C.); tahcalvino@yahoo.com (C.T.F.); ekombangaris@gmail.com (A.H.M.E.); mbumbancy88@yahoo.com (C.M.M.); 2Department of Biochemistry, Faculty of Science, University of Yaoundé I, Messa, Yaoundé P.O. Box 3851, Cameroon; 3Department of Biochemistry, Faculty of Medicine and Biomedical Science, University of Yaoundé I, Messa, Yaoundé P.O. Box 3851, Cameroon; 4Institute of Medicine and Medicinal Plants Studies, Dschang 00237, Cameroon; 5Department of Biochemistry, Faculty of Science, University of Dschang, Dschang 00237, Cameroon; 6Department of Biochemistry, Faculty of Science, University of Buea, Buea P.O. Box 63, Cameroon; 7Institut Universitaire de Technologie, University of Ngoundere, Ngoundere BP 61207, Cameroon; 8Department of Microbiology, Faculty of Science, University of Yaoundé I, Yaoundé P.O. Box 3851, Cameroon

**Keywords:** asymptomatic malaria, children prevalence, *Plasmodium* species, Yaoundé, Cameroon

## Abstract

Malaria is still a threat to public health as it remains the first endemic disease in the world. It is a pervasive parasitic disease in tropical and subtropical regions where asymptomatic malaria infection among humans serves as a significant reservoir for transmission. A rapid and correct diagnosis is considered to be an important strategy in the control of the disease especially in children, who are the most vulnerable group. This study assessed the prevalence of asymptomatic malaria in children at the Nkolbisson health area in Yaoundé, Cameroon. A cross-sectional study design and a convenience sampling plan were used. A total of 127 participants were recruited after informed and signed consent from parents and/or guardians. Blood samples were collected by finger-pricking and venipuncture from children aged 6 months to 10 years and then screened for asymptomatic parasitemia by a rapid diagnostic test (RDT), light microscopy (LM) staining with Giemsa and 18S rRNA polymerase chain reaction (PCR) for speciation. The data were analyzed using SPSS version 20 software. The study identified 85 children who were positive from the PCR, 95 positive from the RDT and 71 from the LM, revealing a malaria prevalence of 66.9%, 74.8% and 55.9%, respectively. The prevalence was not observed to be dependent on the sex and age group of the participants. *Plasmodium falciparum* was the predominant species followed by *Plasmodium malariae* and then *Plasmodium ovale*. The RDT and LM had the same sensitivity (90.6%) with a slight difference in their specificity (RDT: 57.1%; LM: 54.8%). The RDT also demonstrated higher positive and negative predictive values compared with those of the LM.

## 1. Introduction

Malaria is still a threat to public health as it remains the first endemic disease in the world. It is a pervasive parasitic disease in tropical and subtropical regions. It is mostly prevalent in Sub-Saharan Africa, Asia and Latin America [1,2]. The Sub-Sahara African region accounts for the greatest burden in terms of morbidity and mortality. In 2019, the WHO African region accounted for 94% of both the 229 million malaria cases and the 409,000 malaria deaths reported globally [3]. The disease is caused by parasites belonging to the Plasmodium genus, which are transmitted to humans through the bite of infected female Anopheles mosquitoes. Five *Plasmodium* species have been identified as causing malaria with *Plasmodium falciparum* accounting for the greater proportion of the global malaria burden especially in children under the age of 5 years [4]. Malaria has been a hindrance to national growth in almost all African countries to date as it affects major life areas such as education and agriculture and especially the workplace [5]. A *Plasmodium falciparum* infection can result in a disease manifested by the asymptomatic carriage of uncomplicated or severe malaria [6,7]. Asymptomatic malaria refers to the presence of malaria parasites in the blood without symptoms, which usually provides a reservoir for transmission and is an antecedent to symptomatic malaria [8]. About 24% of the population in high transmission settings are affected by widespread asymptomatic parasitemia [4,7]. In malaria-endemic areas, the manifestation of malaria cases is a function of immunity with respect to age and exposure to the parasite; reasons why asymptomatic malaria is common among older children and adults in these areas [6]. Most asymptomatic malaria investigations have been conducted among individuals within this age group living with children under the age of 5 years. Asymptomatic malaria remains a major hindrance to malaria elimination as asymptomatic infections go undetected and untreated with little or no clinical manifestation. These individuals then serve as reservoirs for mosquito infections [9]. The study of asymptomatic malaria cases in children has been given little attention in prevention and control programs. In Cameroon, the management of malaria by the National Malaria Control Program (NMCP) is based on the detection of parasites in the sick followed by a treatment with an artemisinin-based combination therapy (ACT) [10]. Evidence has shown that asymptomatic parasitemia can impair cognition and cause anemia in the host as well as being a good source of the parasite [11]. Understanding the malaria burden in children has a great implication in the interruption of malaria transmission. Ultimately, an accurate, prompt and cheap diagnosis of an infectious disease such as malaria is a key step in managing and treating the disease. Monitoring changes in the prevalence rates is useful for measuring the outcomes of antimalarial interventions especially in the most vulnerable groups such as children. This is important in informing policy makers regarding the optimal strategies to reduce the malaria burden [12]. To complement the insufficiency of data in the prevalence of asymptomatic malaria in Cameroon especially in vulnerable groups such as children and to address the problem of almost a complete absence of data on asymptomatic malaria in swampy areas such as Nkolbisson in Cameroon, this work reports the results of a survey that used an RDT, microscopy and a PCR to assess the asymptomatic carriage of *Plasmodium falciparum* and non-*falciparum* species in children resident in Nkolbisson, Yaoundé, Cameroon.

## 2. Materials and Methods

### 2.1. Study Site

The study was carried out in the Nkolbisson health area in the Cite-Vert health district in the Centre region of Cameroon where Yaoundé is the capital city. Yaoundé has a population of about 2,500,000 inhabitants and covers a surface are of 183 km^2^. Yaoundé is located 700 m (2300 feet) above sea level and at 3′52′ N 11°31′ E. The climate is of a Guinean-type with two rainy seasons (March–June and September–November) and two dry seasons (December–February and July–August). The average annual rain fall here is 1643 mm with a relative humidity ranging from 85 to 90% [13,14,15]. Malaria transmission here is year-round with a peak transmission observed during the rainy seasons [16].

### 2.2. Ethical Consideration

The study received ethical approval from the National Ethics Committee for Human Health Research (N° 2018/07/1091/CE/CNERSH/SP). Research authorizations were also obtained from the Centre Regional Delegation for Public Health and from the health centers where the study was conducted. Prior to the participant enrollment, written and signed consent from each parent or guardian was obtained based on the informed consent forms designed for the study. The potential risks and benefits as well data privacy and confidentiality were clearly explained to parents or guardians. Only children whose parents or guardians signed and returned the consent form were included in the study.

### 2.3. Study Design

This study was a cross-sectional study and a convenience sampling approach was used. All children resident in the study area aged 6 months to 10 years were eligible. All participants were examined by a physician for malaria. Children who presented signs and symptoms of malaria such as a headache, tiredness or fever were not enrolled. Finger-pricking for blood samples was performed on children whose parents or guardians consented to the study. A questionnaire was used to collect the demographic data and vital signs as well as to assess the knowledge and attitude of the participants toward malaria prevention and control. A total of 127 participants were recruited between November 2019 and February 2020.

### 2.4. Malaria Screening and Blood Sampling

Venous blood samples were collected from each patient by a syringe. A rapid diagnostic test (RDT) was performed on the spot in the field using a malaria rapid diagnostic kit supplied by SD Bioine (batch number: 05EDD011B) for the National Malaria Control Program in Cameroon to detect two antigens of the malaria parasites (HRP2/pLDH) that were at the different health centers, as described by the WHO [17]. About 6 µL of the blood was used to prepare thick blood smears. The rest of the blood was placed in EDTA tubes. The EDTA tubes alongside the dried thick blood smears were transported to the laboratory. The blood samples were stored at −20 °C for a further analysis and the slides were stored at room temperature. The thick blood smear was stained with 10% Giemsa stain and read at a 100 X oil immersion objective of the microscope. Both the asexual and sexual stages of the parasites were assessed in the blood smear by comparing the ratio of infected red blood cells with the uninfected ones. The parasites were counted against 200 leucocytes and expressed as parasites/microliter (μL) of blood. As quality control measures, two microscopists examined the slides independently whilst a third microscopist reexamined any slides with a discordance. Samples with a discrepancy between the microscopy results and the results of the *Plasmodium* species-specific PCR were blindly submitted to a second independent microscopist. 

### 2.5. DNA Extraction

Genomic deoxyribonucleic acid (DNA) was extracted from 200 mL of blood in EDTA tubes using an EZNA DNA extraction blood kit (Omega BIO-TEK, Norcross, GA, USA) according to the manufacturer’s instructions. Prior to the extraction, a few materials were sterilized by autoclaving and the equipment was disinfected with 10% bleach and 70% alcohol. The blood samples were briefly thawed at room temperature. In a sterile Eppendorf tube, 100 µL blood was introduced with the help of a micropipette. A total of 150 µL of elution buffer (65 °C) was then added onto it. An OB protease and a BL buffer (lysis buffer) was then added to the mixture and vortexed for 15 s, after which it was incubated at 65 °C for 10 min. During the incubation, the tubes were briefly vortexed twice. A total of 260 µL of 100% ethanol was then added to the mixture, vortexed for 20 s and then centrifuged briefly. The mixture was then transferred to a DNA spin column (DNA HiBind Column) and centrifuged for 1 min at 10,000 rpm. A total of 700 μL of a wash buffer was then introduced into each column. This step was repeated for each tube. A 2 min centrifugation at 10,000 rpm was performed to dry the column matrix. The columns were transferred into a nuclease-free microcentrifuge tube. A total of 150 µL of elution buffer was used to elute the DNA, which was then stored at −20 °C for a further analysis.

### 2.6. Amplification of DNA

A PCR was performed on all samples and the 18S rRNA gene was used to detect and identify the different malaria-causing species. The PCR amplification of the *Plasmodium* species was performed in accordance with a protocol described by Putapomtip et al. [18,19]. A nested PCR was used to amplify the gene. For the first nested amplification reaction, a 25 µL reaction mixture was used; each contained 3 µL of DNA extract, 18.25 µL of nuclease-free water, 2.5 µL of 10 X thermopol buffer (New England Biolabs, Ipswich, MA, USA), 0.5 μL of 10 mM dNTPs, 0.25 μL of *Plasmodium* genus-specific outer primers (Table 1) and 0.25 μL of Taq polymerase (New England Biolabs). The pre-denaturation was performed at 95 °C for 15 min followed by 25 cycles of denaturation at 58 °C for 2 min, annealing at 72 °C for 5 min and an extension at 94 °C for 1 min with a final extension at 72 °C for 5 min. One microliter (1 μL) of the PCR product from the primary PCR was used as a DNA template for the secondary PCR, in which the amplification of the 18S rRNA for *P. falciparum*, *P. malariae* and *P. ovale* was performed in separate reaction tubes. The amplification reaction and thermal cycling profile for the secondary PCR were essentially the same as those for the primary PCR except for a respective pair of the species-specific inner primers (rFAL1/rFAL2, rOVA1/rOVA2, rMAL1/rMAL2). A volume of 25 µL from the first reaction contained 20.25 µL of nuclease-free water, 2.5 µL of 10 X thermopol buffer (New England Biolabs), 0.5 µL of 10 mM dNTPs, 0.25 µL of primer, 0.25 µL of Taq polymerase (New England Biolabs) and 1 μL of amplicons. This second step took place under the following conditions: 95 °C for 15 min, 30 cycles of 58 °C for 2 min, 72 °C for 5 min and 94 °C for 1 min followed by 58 °C for 2 min and 72 °C for 5 min.

## 3. Results

### 3.1. The Characteristics of the Study Population

A total of 127 children were enrolled into the study (Table 2). There was a greater number of males enrolled in the study compared with females. Of the 127 participants recruited, 59/127 (46%) were females and 68/127 (54%) were male. The majority of the children (69/127 (54.3%)) belonged to the age group 5–10 years inclusive and the rest 58/127 (45.7%) were 6 months and 5 years of age. The mean age of the study population was 5.42 ± 2.64. Variations in temperature were observed between 36 °C and 39.9 °C with an overall average of 37.16 °C ± 0.63 °C.

### 3.2. Malaria Prevalence by LM, RDT and PCR Methods

The three diagnostic methods, light microscopy (LM), a rapid diagnostic test (RDT) and a polymerase chain reaction (PCR), detected significant malaria infections in asymptomatic children. A total of 95 (74.8%) from the 127 cases detected positive for a *Plasmodium falciparum* malaria infection by the RDT, 85 (66.9%) by light microscopy and 71 (55.9%), by the PCR. From the patients screened by the PCR, 66.9% were positive, among which were 51 (75.0%) males and 34 (57.6%) females. A total of 42 (33.1%) children were negative for malaria by the PCR. Of the 95 (74.8%) children diagnosed positive for *Plasmodium falciparum* malaria by the RDT, 54 (79.4%) were males and 41 (69.5%) females. In addition, 32 (25.2%) of the children were diagnosed negative for *Plasmodium falciparum* malaria by the RDT, among which were 18 (30.5%) females and 14 (20.6%) males. When the children were diagnosed for *Plasmodium* malaria using microscopy, 71 (55.9%) of them were positive and 56 (44.1) negative. Among the positive cases 17 (28.8%) were females and 54 (79.4%) were males whereas among the negative cases, 42 (71.2%) were females and 14 (20.6%) were males (Table 3). A high prevalence of asymptomatic malaria was observed in the male children for all the diagnostic methods compared with their female counterparts. Asymptomatic malaria infections were found to be high in children between the ages of 5 and 10 years compared with those below the age of 5 years (Table 3). No significant association was observed between sex and the prevalence of malaria as diagnosed with the RDT (*p* = 0.199) and microscopy (*p* = 0.282). However, an association was observed between sex and malaria prevalence by the PCR (*p* = 0.038). No significant association was observed between the age group and malaria prevalence, be it by the RDT (*p* = 0.508), microscopy (*p* = 0.682) or by the PCR (*p* = 0.655).

### 3.3. Prevalence of Plasmodium spp. Infections in Children

In this study, 85 of the infected patients had *Plasmodium falciparum*, 4 were infected with *Plasmodium ovale* and 7 were infected with *Plasmodium malariae*. Of the 85 infected with *Plasmodium falciparum*, 51 (75.0%) were males and 34 (57.6%) were females. A statistical significance was observed between the gender of the children and infection by *Plasmodium falciparum* (*p* = 0.038) (Table 4). Of the 4 patients infected with *Plasmodium ovale*, only 1 (1.7%) was female and 3 (4.4%) were males. Furthermore, 4 (5.9%) of the 7 children infected by *Plasmodium malariae* were males and 3 (5.1%) were females. No statistical significance was observed between the gender in children infected with *Plasmodium ovale* (*p* = 0.382) as well as *Plasmodium malariae* (*p* = 0.844) (Table 4).

### 3.4. Diagnostic Test Performance Characteristics

Overall, when light microscopy (LM) and RDT diagnostic tools were compared with the PCR as the standard, 77 out of the 85 children that were diagnosed positive for malaria by the PCR were also positive by microscopy, giving a sensitivity or true positive rate of 90.6% with a false negative rate of 9.4% (Table 5). Of the 42 children negative for malaria by the PCR, 23 were both negative for the PCR and microsco py, giving a true negative rate or specificity of 54.8% with 45.25% as the false positive rate. In the case of the RDT, a similar trend was observed with the sensitivity and false negative rate as the microscopy. The specificity for the RDT stood at 57.1% with a false positive rate of 42.9%. The probability that children who were diagnosed positive for *Plasmodium falciparum* malaria by microscopy and who were truly infected with *Plasmodium falciparum* was 80.2% and that of those who were diagnosed negative and who were truly negative was 74.2%. These were the respective positive and negative predictive values for microscopy. The positive and negative predictive values of the RDT stood at 81.1% and 75.0%, respectively.

### 3.5. Density of the *Plasmodium* Parasite

The average parasitemia of each child was 8260.19 ± 20,427.84 parasites/µL (Table 6). Among the children diagnosed for malaria by microscopy, 35 (27.6%) had a parasitemia level of <500 parasites/µL, 88 (69.3%) with a parasitemia level greater than 1000 parasites/µL and only 4 (3.1%) between 500 and 1000 parasites/µL. There was no statistically significant association between the parasitemia level and sex (*p* = 0.532) or between the age groups of the children (*p* = 0.698).

## 4. Discussion

This study assessed the prevalence of asymptomatic malaria in children in Nkolbisson, Yaoundé. Nkolbisson is in a forest malaria transmission geo-ecological zone where malaria transmission is year-round [20]. The contributive transmission of malaria in this area is due to its topographic nature characterized by many swamps, which are breeding grounds for mosquitoes. Asymptomatic malaria patients serve as reservoirs for malaria transmission [21]. Even though asymptomatic malaria may be beneficial in inducing and sustaining a partial immunity against malaria, it may give rise to severe disease complications characterized by a coma, acidosis or severe anemia [22]. The aim of this study was to assess the prevalence of the asymptomatic carriage of *Plasmodium falciparum* and non-*falciparum* species in children resident in Nkolbisson, Yaoundé, Cameroon. LM, RDT and PCR malaria diagnostic techniques were used to detect *Plasmodium falciparum* and non-*falciparum* species in children between the ages of 6 months and 10 years. The study compared the diagnosis of malaria by LM, a RDT and a PCR with the assessment of asymptomatic malaria in children. A total of 127 asymptomatic children were tested for malaria. The study identified 85 children as positive by the PCR, 95 positive by the RDT and 71 by LM, revealing a malaria prevalence of 66.9%, 74.8% and 55.9%, respectively.

Based on LM (55.9%) and the RDT (74.8%), the study reported a high prevalence of asymptomatic malaria in children. This prevalence was higher compared with that observed in other studies carried out in Cameroon; for instance, in Buea where a prevalence of 40.32% from LM and 34.41% from an RDT was observed and in Yaoundé with an overall prevalence of 6.5% [23,24]. A similar prevalence was again observed in Buea nine years subsequently but by following the asymptomatic screening of blood donors [25]. The great difference in the prevalence of our study and that in Yaoundé may be due to the difference in seasons. Our study was performed in the rainy seasons when transmission is higher compared with the dry seasons when that of Yaoundé was conducted. The prevalence was also higher when compared with other studies conducted in children not in Cameroon. In Ghana, a prevalence of LM of 23.2% and an RDT of 31.2% was observed [26]. This difference may be explained by the fact that the study in Ghana was performed on children under the age of 5 years whereas this study was performed on children aged between 6 months and 10 years; therefore, there could have been a possible gain in protective immunity in the older children because malarial protective immunity is acquired over several years [27]. This prevalence was also higher than others reported in India [28], Dakar [29], Tanzania [30] and Pakistan [31]. 

Based on the PCR, a high prevalence of 75.0% was observed compared with a similar study carried out in Ghana where a prevalence of 36.8% of asymptomatic *Plasmodium* infections was observed [26]. A similar prevalence was observed in a study conducted by Dinko et al. [32] with a prevalence of 76.6%. This similarity can be explained by the fact that they included not only children below 5 years of age but also children above five years of age. Older children are major carriers of asymptomatic *Plasmodium* compared with younger children [27]. Overall, the prevalence of asymptomatic malaria was found to be greater in males than in females in both the PCR, LM and the RDT. A significant distribution among males and females was observed in the case of the PCR. This was similar to that obtained in Ethiopia [21]. The trend in the prevalence of asymptomatic malaria was found to increase with the age group. This showed that individuals developed immunity against malaria with a repeated exposure to mosquito bites [33].

The distribution of the *Plasmodium* carriage showed no significant association with gender in the case of *Plasmodium malariae* (*p* = 0.382) but a significant association was observed with *Plasmodium falciparum* (*p* = 0.038) and *P. ovale* (*p* = 0.382). The total number of species identified was 96, which was slightly than the 95 positive sample diagnoses by the RDT. This difference is due to the fact that one individual was found to be harboring more than one species. The average parasitemia was not significantly associated with the prevalence of parasitemia in either gender or the age group. However, other studies [34,35] have indicated that there are associations with the two variables [36]. The sensitivity and specificity of the different tests were determined using venous blood and the PCR was used as the reference assay. Of the two malaria diagnostic assays evaluated, the RDT demonstrated the highest specificity (57.1%) and negative predictive value (NPV) (75.0%). This indicated that the probability that the RDT correctly identified children without the disease was 57.1% and the probability that a child was diagnosed negative when he or she was truly negative was 75.0%. The sensitivity of the RDT and LM were the same (90.6%) with LM recording the highest positive predictive value. The sensitivity and specificity of the RDT and LM were similar to a study conducted in Cameroon and Tanzania. This similarity can be accounted for by the fact that this study was conducted in the south-west region of Cameroon, which falls within the same transmission zone as our study site and by the fact that Cameroon and Tanzania are malaria-endemic countries [1,20,37]. The specificity of the RDT was lower than that reported by another study in southern Nigeria. This difference may be due to the high false positive rate of the RDT compared with the PCR [2]. The predictive values obtained in this study, apart from the LM PPVs, were in line with another study [37].

## 5. Conclusions

Our study identified 85 children positive by a PCR, 95 children positive by an RDT and 71 by LM, revealing a malaria prevalence of 66.9%, 74.8% and 55.9%, respectively. *Plasmodium falciparum* was the predominant species followed by *Plasmodium malariae* and then *Plasmodium ovale*. Thus, understanding the malaria burden in children has a great implication on the interruption of malaria transmission. Ultimately, an accurate, prompt and cheap diagnosis is a key step in managing and treating the disease.

## Figures and Tables

**Table 1 children-08-00960-t001:** Primers used in this study.

No	Species	Primer Name	Primer Sequences (5′–3′)	Amplicon Size
1	*Plasmodium* spp.	rPLU5	CCTGTTGTTGCCTTAAACTTC	1200
		rPLU6	TTAAAATTGTTGCAGTTAAAACG	
2	*P. falciparum*	rFAL1	TTAAACTGGTTTGGGAAAACCAAATATATT	205
		rFAL2	ACACAATGAACTCAATCATGACTACCCGTC	
3	*P. ovale*	rOVA1	ATCTCTTTTGCTATTTTTTAGTATTGGAGA	880
		rOVA2	GGAAAAGGACACATTAATTGTATCCTAGTG	
4	*P. malariae*	rMAL1	ATAACATAGTTGTACGTTAAGAATAACCGC	144
		rMAL2	AAAATTCCCATGCATAAAAAATTATACAAA	

P: *Plasmodium*; spp: species; r: ribosomal; No: number.

**Table 2 children-08-00960-t002:** Characteristics of the study population.

Characteristics	*n*		Chi-Squared (X^2^)	*p*-Value
Gender	M	68 (54%)	4.308	0.038
	F	59 (46%)		
Age (years)	<5	58 (45.7%)	0.2	0.655
	5–10	69 (54.3%)		
Temperature (°C)	37.16 °C ± 0.63 °C (36–39.9 °C)			
Mean age (years)	5.42 ± 2.64			
Mean parasitemia/µL	8260.19 ± 20,427.84			

M: male; F: female; *n*: number of participants.

**Table 3 children-08-00960-t003:** Prevalence of asymptomatic malaria in children by a PCR, LM and an RDT by gender and age group.

	PCR Results					RDT Results					Microscopy Results				
**Parameter**	*n*	Prevalence of malaria % (*n*)		(X^2^)	*p*	*n*	Prevalence of malaria % (*n*)		(X^2^)	*p*	*n*	Prevalence of malaria % (*n*)		X^2^	*p*
		Positive	Negative				Positive	Negative				Positive	Negative		
**Gender**															
**Male**	68	75.0% (51)	25.0% (17)	4.308	0.038	68	79.4% (54)	20.6% (14)	1.649	0.199	68	79.4% (54)	20.6% (14)	1.158	0.282
**Female**	59	57.6% (34)	42.4% (25)			59	69.5% (41)	30.5% (18)			59	28.8% (17)	71.2% (42)		
**Total**	127	66.9% (85)	33.1% (42)			127	74.8% (95)	25.2% (32)			127	55.9% (71)	44.1% (56)		
**Age (yrs)**															
**<5**	58	47.1% (40)	42.9% (18)	0.2	0.655	58	72.5% (50)	22.4% (13)	0.439	0.508	58	77.6% (45)	22.4% (13)	0.631	0.682
**5–10**	69	52.9% (45)	24 (57.1%)			69	77.6% (45)	27.5% (19)			69	73.9% (51)	26.1% (18)		

A statistical significance was observed at *p* < 0.05. *n*: number of participants; X^2^: Pearson chi-squared test; *p* = *p*-value; yrs: years; %: percentage of participants.

**Table 4 children-08-00960-t004:** Association of the prevalence of asymptomatic malaria species with gender.

Parasite Species		Gender	Total	X^2^	*p*
	Male (*n* = 68)	Female (*n* = 59)			
Plasmodium falciparum	75.0% (51)	57.6% (34)	85	4.308	0.038
Plasmodium ovale	4.4% (3)	1.7% (1)	4	0.764	0.382
Plasmodium malariae	5.9% (4)	5.1% (3)	7	0.039	0.844

A statistical significance was observed at *p* < 0.05. *n*: number of participants; X^2^: Pearson chi-squared test; *p*: *p*-value.

**Table 5 children-08-00960-t005:** Sensitivity and specificity and negative and positive predictive values of LM and the RDT (with the PCR as a reference method).

	RDT			Microscopy		
**Statistics**	Value	95% CI		Value	95% CI	
**Sensitivity**	90.6%	0.6654	0.8390	90.6%	0.6731	0.8458
**Specificity**	57.1%	0.1509	0.5110	54.8%	0.1410	0.5034
**Positive predictive value**	81.1%	0.7144	0.8809	80.2%	0.7057	0.8737
**Negative predictive value**	75.0%	0.1212	0.4375	74.2%	0.1253	0.4492
**Positive likelihood ratio**	110.1%	0.8336	1.4546	109.4%	0.8377	1.4293
**Negative likelihood ratio**	77.2%	0.4596	1.2976	77.63%	0.4564	1.3201
**Disease prevalence**	79.53%	0.7126	0.8596	78.74%	0.7040	0.8529

RDT: rapid diagnostic test; CI: confidence interval.

**Table 6 children-08-00960-t006:** Distribution of the parasite density in a microscopic study within gender and age.

Characteristics	Number of Parasites/µL			Total	X^2^	*p*
	<500	500–1000	>1000			
**Sex**						
**Male**	23.5% (16)	2.9% (2)	73.5% (50)	68	1.262	0.532
**Female**	32.2% (19)	3.4% (2)	64.4% (38)	59		
**Total**	27.6% (35)	3.1% (4)	69.3% (88)	127		
**Age**						
**<5 years**	27.6% (16)	1.7% (1)	70.7% (41)	58	0.719	0.698
**5–10 years**	27.5% (19)	4.3% (3)	68.1% (47)	69		
**Total**	27.6% (35)	3.1% (4)	69.3% (88)	127		

A statistical significance was observed at *p* < 0.05. *p*: *p*-value; X^2^: Pearson chi-squared test.

## Data Availability

No applicable.

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
