# Peer review of "Assessing Asymptomatic Malaria Carriage of Plasmodium falciparum and Non-falciparum Species in Children Resident in Nkolbisson, Yaoundé, Cameroon"

_children, 2021, doi:10.3390/children8110960_

Round 1
Reviewer 1 Report
I have gone through this paper and I have seen it is well presented and written. Before its acceptance, I suggest that the following points can be revised:
1. The Introduction section needs a major revision in terms of providing a more accurate and informative literature review and the pros and cons of the available approaches and how the proposed method is different comparatively. Also, the motivation and contribution should be stated more clearly, as well as the organization of the paper must be clearly established
2. The authors must explain the contribution of this manuscript over and above the state of art and establish its novelty.
Author Response
Dear reviewer please see the attachment bellow

Reviewer 2 Report
Reviewer comments:
General comments
The submitted manuscript assessed the prevalence of asymptomatic malaria in children at the Nkolbisson health area in Yaoundé Cameroon. A total of 127 participants were recruited in this cross-sectional study. Blood samples were collected by finger-pricking and venipuncture from children aged 6 months to 10 years and then screened for asymptomatic parasitemia by Rapid Diagnostic Test (RDT), light microscopy (LM) staining with Giemsa and 18S rRNA Polymerase Chain Reaction (PCR) for speciation. Based on the results, the study identified 85 children positive by PCR, 95 positive by RDT and 71 by LM, revealing a malaria prevalence of 66.9%, 74.8% and 55.9% respectively.
Overall, the text is easy to follow, clear, and is properly discussed with other studies that have been published. However, there are some minor issues in the manuscript that need to be addressed before publication.
Minor comments
I would like to include some comments and suggestions for the authors, so they can judge or consider to refer to them during the revisions:
Lines 31-32: Please add the positivity index for each species.
Line 79: Add when the study was performed. Months...years?
Line 102: Please add more details about the questionnaire. This information is relevant to readers.
Line 105: Please insert more details about the RDT manufacturer.
Line 107: Please check this sentence “different health centers as described by…” Described by?
Line 121: Please insert more details about the DNA extraction kit manufacturer.
Lines 138-139: Please check this sentence “Putapomtip et al [18] and [19]?
Line 139: Clarify what the target is.
Line 145: 1 μL, right? Please check.
Line 149: Please remove italic from PCR.
Table 1 could be moved to supplementary information.
Line 169: PCR right? Instead qPCR.
Lines 217-219: A total of 95 (74.8%) out of the 127 case were detected positive for Plasmodium falciparum malaria infection by RDT, 85 (66.9%) by light microscopy and 71 (55.9 %,) by PCR. But when evaluating the species, the authors reported a total of 96 samples. What are the reasons for this finding? Could the authors clarify this?
Table 4: Please add italic for plasmodium species.
Table 5: Please provide the values for other parameters: prevalence and overall accuracy. Moreover, please add the 95% confidence interval (CI) for each parameter. Authors can use this software and this article as an example to improve the presentation of Table 5.
https://www.nature.com/articles/s41598-021-83371-1
https://www.medcalc.org/calc/diagnostic_test.php
Lines 230-241: In my opinion, this session is confusing. I suggest rewriting after modifying table 5. To rewrite: Please consider the results as: true positive, true negative, false positive, and false negative.
Lines 255-270: Most of this paragraph has already been described earlier in the manuscript. I believe authors need to be more direct in the discussion.
Lines 287-297: I would like to see more discussion of high prevalence by gender and age… The authors could further compare their results with findings from other studies.
Discussion: I believe the authors can also use the discussion to discuss the importance of developing faster tests to monitor the presence of Plasmodium in asymptomatic individuals. Test examples developed for Plasmodium detection? Articles with test validation using patient samples?
Some grammatical errors/typos throughout the manuscript should still be corrected.
Author Response
Dear reviewer,
Thanks for contributing for the betterment of our manuscript.
These has also been amended on the manuscript as well.
Lines 31-32: Please add the positivity index for each species.
Please is it species diversity index. That’s if you don’t mind
Line 79: Add when the study was performed. Months...years?
Participants were recruited from November 2019 to February 2020 this was then followed by a 3 months sample processing and analysis
Line 102: Please add more details about the questionnaire. This information is relevant to readers.
A questionnaire was used to collect demographic data, vital signs and to assess participants’ knowledge and attitude towards malaria prevention and control.
Line 105: Please insert more details about the RDT manufacturer.
Rapid diagnostic kit used was supplied by SD bioline (batch number:05EDD011B) to the National Malaria Control Program in Cameroon to detect two antigens of the malaria parasites (HRP2/pLDH),
Line 107: Please check this sentence “different health centers as described by…” Described by?
Rectified already
Line 121: Please insert more details about the DNA extraction kit manufacturer.
EZNA DNA extraction blood kit (Omega BIO-TEK)
Lines 138-139: Please check this sentence “Putapomtip et al [18] and [19]?
Line 107: Please check this sentence “different health centers as described by…” Described by?
Rectified
Line 139: Clarify what the target is
18S rRNA gene
Line 145: 1 μL, right? Please check
Rectified
Line 149: Please remove italic from PCR.
Done
Table 1 could be moved to supplementary information
Ok
Line 169: PCR right? Instead qPCR
Rectified
Lines 217-219: A total of 95 (74.8%) out of the 127 case were detected positive for Plasmodium falciparum malaria infection by RDT, 85 (66.9%) by light microscopy and 71 (55.9 %,) by PCR. But when evaluating the species, the authors reported a total of 96 samples. What are the reasons for this finding? Could the authors clarify this?
This section is not reporting the number of samples but the number of species within the study population. For instance the difference is due to the fact that one of the participant was harboring two different species.
Table 4: Please add italic for plasmodium species.
done
able 5: Please provide the values for other parameters: prevalence and overall accuracy. Moreover, please add the 95% confidence interval (CI) for each parameter. Authors can use this software and this article as an example to improve the presentation of Table 5.
|
|
RDT |
Microscopy |
||||
|
Statistics |
value |
95% CI |
value |
95% CI |
||
|
Sensitivity |
90.6% |
0.6654 |
0.8390 |
90.6% |
0.6731 |
0.8458 |
|
Specificity |
57.1% |
0.1509 |
0.5110 |
54.8% |
0.1410 |
0.5034 |
|
Positive predictive value |
81.1% |
0.7144 |
0.8809 |
80.2% |
0.7057 |
0.8737 |
|
Negative predictive value |
75.0% |
0.1212 |
0.4375 |
74.2% |
0.1253 |
0.4492 |
|
Positive likelihood ratio |
110.1% |
0.8336 |
1.4546 |
109.4 |
0.8377 |
1.4293 |
|
Negative likelihood ratio |
77.2% |
0.4596 |
1.2976 |
77.63 |
0.4564 |
1.3201 |
|
Disease prevalence |
79.53 |
0.7126 |
0.8596 |
78.74 |
0.7040 |
0.8529 |
Lines 230-241: In my opinion, this session is confusing. I suggest rewriting after modifying table 5. To rewrite: Please consider the results as: true positive, true negative, false positive, and false negative.
Overall, when light microscopy (LM) and RDT diagnostic tools were compared with PCR as the standard, 77 out of the 85 children that were diagnosed positive for malaria by PCR were also positive by microscopy giving a sensitivity or true positive rate of 90.6% with a false negative rate of 9.4%. Of the 42 children negative for malaria by PCR, 23 were both negative for PCR and microscopy giving a true negative rate or specificity of 54.8% and with 45.25% as the false positive rate. In the case of RDT, a similar trend was observed with sensitivity and false negative rate as in microscopy. Indeed, the specificity for the RDT stood at 57.1% with a false positive rate of 42.9%. The probability that children who were diagnosed positive for Plasmodium falciparum malaria by microscopy and that were truly infected with Plasmodium falciparum was 80.2% and that of those who were diagnosed negative and were truly negative was 74.2%. That is the respective positive and negative predictive value for microscopy. The positive and negative predictive values of the RDT stood at 81.1% and 75.0% respectively.
Lines 255-270: Most of this paragraph has already been described earlier in the manuscript. I believe authors need to be more direct in the discussion.
Ok noted with complement. The idea was just to call the attention of the reader on the objectives of the study
Lines 287-297: I would like to see more discussion of high prevalence by gender and age… The authors could further compare their results with findings from other studies.
We tried getting enough publications on this aspect for a prober study design and to better discuss our findings but unfortunately we could not lay hands on enough publications regarding this particular aspect
Discussion: I believe the authors can also use the discussion to discuss the importance of developing faster tests to monitor the presence of Plasmodium in asymptomatic individuals. Test examples developed for Plasmodium detection? Articles with test validation using patient samples?
Thanks a lot for this. Please we promised to address this aspect in our subsequent investigations
Some grammatical errors/typos throughout the manuscript should still be corrected and revised.
Round 2
Reviewer 2 Report
I thank the authors for sharing an improved and revised manuscript.